# Eating the Enemy: Mycoplasma Strategies to Evade Neutrophil Extracellular Traps (NETs) Promoting Bacterial Nucleotides Uptake and Inflammatory Damage

**DOI:** 10.3390/ijms232315030

**Published:** 2022-11-30

**Authors:** Carla Cacciotto, Alberto Alberti

**Affiliations:** 1Dipartimento di Medicina Veterinaria, Università degli Studi di Sassari, 07100 Sassari, Italy; 2Mediterranean Center for Disease Control, Università degli Studi di Sassari, 07100 Sassari, Italy

**Keywords:** neutrophil, innate immunity, neutrophil extracellular traps (NETs), mycoplasma, nuclease, virulence, chronic disease

## Abstract

Neutrophils are effector cells involved in the innate immune response against infection; they kill infectious agents in the intracellular compartment (phagocytosis) or in the extracellular milieu (degranulation). Moreover, neutrophils release neutrophil extracellular traps (NETs), complex structures composed of a scaffold of decondensed DNA associated with histones and antimicrobial compounds; NETs entrap infectious agents, preventing their spread and promoting their clearance. NET formation is triggered by microbial compounds, but many microorganisms have evolved several strategies for NET evasion. In addition, the dysregulated production of NETs is associated with chronic inflammatory diseases. Mycoplasmas are reduced genome bacteria, able to induce chronic infections with recurrent inflammatory symptoms. Mycoplasmas’ parasitic lifestyle relies on metabolite uptake from the host. Mycoplasmas induce NET release, but their surface or secreted nucleases digest the NETs’ DNA scaffold, allowing them to escape from entrapment and providing essential nucleotide precursors, thus promoting the infection. The presence of *Mycoplasma* species has been associated with chronic inflammatory disorders, such as systemic lupus erythematosus, rheumatoid arthritis, inflammatory bowel disease, Crohn’s disease, and cancer. The persistence of mycoplasma infection and prolonged NET release may contribute to the onset of chronic inflammatory diseases and needs further investigation and insights.

## 1. Introduction

Neutrophil granulocytes are the most abundant circulating immune cells and represent the first line of defense against infection; they are paramount to the innate immune response as major effectors of acute inflammation [1]. Upon early infection, inflammatory and chemotactic cytokines recruit neutrophils from the bloodstream to the infection site where they contribute to pathogen clearance. Traditionally, neutrophils eliminate pathogens by two main mechanisms: (a) pathogen internalization coupled with formation and activation of the phagolysosome (phagocytosis), leading to the intracellular killing of infectious agents, and (b) degranulation and release of the microbicidal molecules, which is responsible for extracellular killing [1,2]. Recently, a novel anti-microbial mechanism that is involved in the pathogen’s clearance in the extracellular compartment was described in neutrophils. Briefly, upon stimulation, neutrophils release Neutrophil Extracellular Traps (NETs), complex web-like structures composed of a scaffold of decondensed DNA strings associated with histones and other proteins and peptides with antimicrobial activity, mainly derived from neutrophil granules, such as elastase, myeloperoxidase, calprotectin, cathelicidins, and defensins [3,4]. The release of NETs can be activated both by endogenous stimuli, such as Damage-Associated Molecular Patterns (DAMPs), and by microbial Pathogen-Associated Molecular Patterns (PAMPs) [4,5].

The main role of NETs is to prevent the spread of infection in the host by entrapping extracellular pathogens [1,3]. Moreover, the release of NETs promotes the direct killing of entrapped pathogens and boosts bacteria phagocytosis by macrophages, acting as a pseudo-opsonin [3,6,7].

In recent years, a large variety of microorganisms have been reported to induce NET formation, such as Gram-positive and Gram-negative bacteria, mycobacteria, mollicutes, fungi, viruses, and protists [3,8,9,10,11,12,13,14,15,16,17]. However, due to selective pressure, many pathogenic microorganisms evolved strategies to overcome NETs. NET evasion mainly relies on the inhibition of NET release, degradation of NET components, and resistance to NET-mediated killing [18]. Among them, the degradation of the NET’s DNA backbone by means of surface or secreted nucleases is the most commonly reported mechanism.

NETs are also implicated in the pathogenesis of acute and chronic inflammatory disorders, such as several autoimmune diseases [19].

Mycoplasmas are the smallest described bacteria, able to infect and cause severe syndromes in a wide range of animal species, including humans. Mycoplasmas are extracellular pathogens and are usually found attached to the surface of host cells, but occasionally they can invade the intracellular compartment [20,21,22,23,24,25,26]. In addition, *Mycoplasma hominis* (*M. hominis*) is able to stably infect and replicate within the protozoan *Trichomonas vaginalis* (*T. vaginalis*) [27].

Mycoplasmas evolved from Gram-positive bacteria through a reductive process, which led to the loss of several gene clusters [21]. Due to this process, mycoplasmas have a reduced metabolic capacity and developed a parasitic lifestyle relying on the metabolites supplied by the host cell, such as nucleotides. This is fundamental to the survival of mycoplasmas, which lost the ability to synthesize de novo purine and pyrimidine bases and depend on the salvage pathway for producing nucleotide precursors [28,29].

Furthermore, mycoplasmas lost the ability to synthesize a bacterial cell wall, lacking many typical bacterial PAMPs associated with the cell surface, such as lipoteichoic acids, flagellin, and some lipopolysaccharides, which makes recognition by the host immune cells more difficult, with implications for the pathogenic process [21,30].

Despite mycoplasmas being recognized as causative agents or co-factors in many diseases causing chronic infections, the implication of the host immune response in the pathogenic processes is still under-investigated [31].

The interplay between mycoplasma–neutrophil granulocytes has been poorly defined. Evidence has demonstrated that mycoplasma viability is not significantly reduced by the interaction with neutrophils, demonstrating a resistance against their killing strategies [32]. As a consequence, the role of neutrophils in mycoplasma infection and the pathogenesis of mycoplasmosis was underestimated. However, after the discovery of NETs, the interplay between mycoplasmas and neutrophils is back in the spotlight. Recently, the ability of mycoplasmas to induce NET formation both in vitro and in vivo was reported, and the key triggering factors were identified. Moreover, resistance to NET killing was demonstrated and was found to be associated with the expression of surface-located or secreted nucleases capable of dismantling NETs [16,33,34,35,36,37,38,39,40].

Here, we critically review data reported in the literature about the involvement of NETs in mycoplasma virulence and pathogenesis, and we outline a mechanism through which mycoplasmas induce and evade NETs while at the same time gathering nucleotides from the host.

## 2. Role of NETs during Bacterial Infection

The formation of NETs is an innate immune mechanism conserved in vertebrates and aimed at fighting microbial infections of various natures [7,41,42,43]. Among microbial agents, bacteria are the most studied, and many species are reported as NETs-inducers, such as *Staphylococcus aureus*, *Streptococcus pneumoniae*, *Haemophilus influenzae*, *Mycobacterium tuberculosis*, *Vibrio cholerae*, and many *Mycoplasma* species [8,16,34,37,39,40,44,45,46,47].

Upon stimulus, neutrophils initiate NET release with the disintegration of the nucleus and granule membranes. As a consequence, decondensed chromatin diffuses into the cytoplasm, coming in contact with cytoplasmic and granule proteins. NET formation is completed with the rupture of the cytoplasmic membrane and the release of the NETs into the extracellular space [48,49,50,51].

Although neutrophils’ ability to release NETs is conserved along vertebrate evolution for its strategic importance in host defense, mechanisms through which NETs contribute to bacteria clearance still need deeper investigation and insights. It is commonly accepted that NETs physically entrap bacteria, blocking them at the site of infection and preventing their spread to other body regions [3]. One of the NET’s direct antimicrobial properties is linked to the presence of DNA that chelates cations present in the extracellular compartment and disrupts microbial cell membranes [52,53].

Due to the massive presence of antimicrobial peptides (AMPs), it was initially supposed that NETs could directly exercise a bactericidal action; however additional evidence has diminished the role of AMPs in the direct killing of entrapped bacteria [54,55]. In fact, elastase and proteases associated with NETs considerably reduce AMPs’ activity, thus suggesting a bacteriostatic rather than bactericidal activity for NETs [55,56,57]. Mostly, NETs carry out their antimicrobial activities indirectly, by activating or enhancing several pathways. It was demonstrated that NETs released by human neutrophils against *Pseudomonas aeruginosa* and *S. aureus* entrap bacteria without directly killing them while promoting complement-mediated killing [58]. Moreover, NETs facilitate the clearance of several extracellular bacteria by acting as pseudo-opsonins, thus enhancing phagocytosis by macrophages. During this process, macrophages internalize neutrophil-derived AMPs into the phagosome where their activity is fully restored [6].

Interestingly, most studies on bacterial survival were either conducted ex vivo or in vitro, in static culture conditions. It was observed that NETs generated in static conditions (such as by adherent neutrophils) have fewer killing abilities than NETs released under dynamic conditions (non-adherent neutrophils). This phenomenon should be investigated more deeply and should be kept into account as experimental conditions could dramatically affect the results of such experiments [58].

Similarly to what was observed in necrosis, NETs induce proinflammatory responses [59,60]. The translocation of NETs’ DNA into macrophages’ cytosol activates the cyclic guanosine monophosphate–adenosine monophosphate synthase (cGAS), inducing the production of type I interferon (IFN) [61]. Moreover, when exposed to NETs, macrophages release IL-1β, IL-6, IL-8, IL-18, and TNFα [62,63,64,65]. Type I IFN is induced in plasmacytoid dendritic cells [66]. On the other side, NETs reduce the LPS-induced secretion of proinflammatory cytokines by macrophages and dendritic cells and the expression of antigen-presenting molecules by dendritic cells, thus exerting anti-inflammatory effects and acting as immunomodulators as well [64,67,68].

As mentioned above, in order to escape NET entrapment, many bacteria, such as *S. aureus*, *Streptococcus* spp., *V. cholerae*, *Neisseria gonorrhoeae*, *Yersinia* spp., and *Mycoplasma* spp., produce surface-exposed or secreted nucleases which dismantle NET’s DNA backbone [16,34,36,39,45,69,70,71]. Furthermore, by digesting and destroying NET structures, nucleases also interrupt the indirect killing pathways, such as the complement-mediated mechanism. In addition, the cooperation between neutrophils and macrophages is compromised, inhibiting the translocation of NET’s AMPs into the macrophage phagosome, and stopping their reactivation.

Indeed, NETs represent a powerful and effective weapon to fight bacterial infections, but their production needs fine-tuning. In fact, an overproduction of NETs can cause tissue damage contributing to the pathogenesis of acute and chronic inflammatory disorders. Moreover, as mentioned above, the massive production of DAMPs triggers NET release, establishing a self-generating process [4,5]. Various data reported in the literature suggest an association between uncontrolled NET formation and non-infectious inflammatory diseases, such as rheumatoid arthritis, allergies, psoriasis, systemic lupus erythematosus, appendicitis, otitis media, preeclampsia, vasculitis, and cancer [62,72,73,74,75]. Therefore, NETs can contribute to the onset and progression of inflammatory diseases through different pathways.

Under physiological conditions, NETs are promptly removed together with pathogens, avoiding prolonged inflammation. One of the described mechanisms involves the digestion by DNase I and the opsonization by C1q, which facilitate phagocytosis of the debris by monocytes-derived macrophages in a process that does not involve the secretion of pro-inflammatory cytokines [76]. In patients with systemic lupus erythematosus, DNase I activity is less relevant, probably due to the presence of anti-DNase I antibodies, increasing the persistence of NETs [51].

Similarly, the presence of constant stimuli, such as bacterial PAMPs and tissue DAMPs, constantly trigger the release of NETs that can be difficult to clear and promote a prolonged inflammatory status. NETs’ presence at the infection site provides a stimulus and a scaffold for platelet and red blood cells adhesion and aggregation, thus promoting coagulation and thrombotic events; moreover, extracellular DNA induces thrombin generation in plasma and increases the protease activity of coagulation factors [77,78,79].

Moreover, damage to NETs’ structures induced by bacterial nucleases causes the release of antimicrobial components that can exacerbate inflammation and lead to host tissue damage and the release of DAMPs [80]. In fact, despite the reduction in their antimicrobial activity in the extracellular compartment, NET-associated AMPs are involved in immunomodulation by acting as strong inflammation promoters [81].

In addition to the DNA scaffold, histones and elastase released upon NETs’ dismantling are also involved in thrombotic events. Free histones can induce epithelial and endothelial cell death, promoting thrombin generation and preventing the thrombomodulin function, while elastase inactivates the tissue factor pathway inhibitor, increasing coagulation and fibrin deposition [51].

Several citrullinated proteins, such as histones, are released with NETs. It has been demonstrated that in patients affected by rheumatoid arthritis, locally released citrullinated histones enhance the generation of highly mutated clonal B cells, producing affinity auto-antibodies and thus exacerbating the inflammatory process [82].

Moreover, according to the American Cancer Society, about 20% of worldwide cancers are triggered by infectious agents. A significant number of epidemiological and molecular studies suggest that infection can trigger cellular transformation either by direct oncogenic properties or indirectly by causing chronic inflammation, as seen in the case of prostate cancer [83]. Dysregulated NET release is involved in cancer progression and the dissemination of metastasis [84]. Indeed, by causing tissue and organ damage, NETs are able to awake dormant cancer cells, entrapping them by providing an adhesion substrate, thus promoting metastasis [85,86,87].

## 3. Mycoplasmas and NETs

The first evidence supporting the hypothesis of NET induction and release by mycoplasmas was provided by recent proteomic investigations on sheep mastitis. Several NET markers, such as citrullinated histones and neutrophil-derived AMPs, were identified in the milk fat globule of sheep infected by *Mycoplasma agalactiae* (*M. agalactiae*) but not in healthy sheep, providing evidence for NETs release during natural infection, in vivo [88,89]. Further studies definitively demonstrated mycoplasmas’ ability to induce NET formation both in vivo and ex vivo [16]. NET’s structures were demonstrated in the mammary gland of ewes infected by *M. agalactiae* by simultaneously co-localizing extracellular DNA, NET-associated proteins, and *M. agalactiae* bacteria clusters; interestingly, in infected tissues, NETs were found mainly in the alveolar lumen, definitively demonstrating the release of these structures in mastitic milk [16,89].

Different PAMPs trigger NET release by activating non-overlapping pathways [90,91,92]. Due to the lack of a bacterial wall mycoplasma, cells display a reduced and simpler PAMP repertoire. Indeed, the mycoplasma cell surface is basically composed of a single lipid bilayer membrane with the lipid-anchored lipoproteins being the most abundant class [21,93,94]. Mycoplasma surface lipoproteins interact with the host cells during infection, fulfilling very diverse functions, including metabolite trafficking, cell adhesion, biofilm formation, cytotoxicity, colonization and invasion of host cells, and immunomodulation [95,96,97,98,99,100]. Among the immunomodulation properties, mycoplasma lipoproteins have been proven to be major determinants of NET release during innate immune response [16]. More specifically, it was experimentally proven that lipopeptides are sufficient to trigger NET release both by ovine and human neutrophils independently of their amino acid sequence, indicating that lipid moieties only are crucial to the induction of NET release [16,34].

Induction of, and escape from NETs by mycoplasmas are summarized in Figure 1.

Upon infection, neutrophils are recruited to the infection site (Figure 1, step 1). Here, neutrophils’ Toll-like receptors (TLRs) recognize the lipopeptide portion of mycoplasma lipoproteins and activate the pathways leading to NET release (Figure 1, steps 2 and 3). Specifically, lipoproteins interact with TLR heterodimers TLR2/6 or TLR1/2, which recognize diacylated or triacylated peptides, respectively, activating the same pathways [101]. In mycoplasmas, the presence of diacylated lipoproteins is well-assessed, while the presence of triacylation has been under debate for a long time [102]. It has been demonstrated that mycoplasma lipopeptides interact with TLR2, inducing the production of IL-8 that promotes NET release and recruits other neutrophils to the site of infection by chemotaxis [16]. Recently, mass spectrometry allowed the identification of a triacylated lipoprotein in *Mycoplasma genitalium* (*M. genitalium*, MG_040), two in *Mycoplasma pneumoniae* (*M. pneumoniae*, MPN052, and MPN415), and two in *Mycoplasma fermentans* (*M. fermentans*, MBIO_0319, and MBIO_0661) [102,103,104]. To date, the enzyme responsible for the third acylation of *Mycoplasma* lipoproteins is still unknown. *Mycoplasma*-derived Pam2cys and Pam3cys peptides are able to induce NET formation in human neutrophils at a comparable level ex vivo [34].

Since the amino acid sequence of lipopeptides does not influence the recognition by TLR2, the mechanisms triggering NET release can be expected to be shared by all *Mycoplasmataceae*. To date, the ability to induce NET release has been observed either in vivo or ex vivo in several mycoplasmas, such as *M. agalactiae*, *M. capricolum* subsp. *capricolum*, *M. hominis*, *M. conjunctivae*, *M. arthritidis*, *M. arginini*, *M. felis*, *M. pneumoniae*, and *M. bovis* [16,34,37,89,105]. Moreover, NET structures have been reported in the amniotic fluid of women with *Ureaplasma* and *Mycoplasma* intra-amniotic infection [106].

Mycoplasmas are not only capable of inducing NETs’ release but can also disassemble them (Figure 1, step 4), both in vivo and ex vivo. The first indications of in vivo NET disassembling were demonstrated by co-localization experiments conducted on mammary gland tissues of ewes infected by *M. agalactiae*. In these tissues, in presence of mycoplasma aggregates, only a few residues of extracellular DNA and histones could be detected, while NETs with well-defined and intact structures could be detected far from the *M. agalactiae* clusters. Degradation of NETs in the presence of mycoplasma aggregates has also been reported ex vivo in both humans and sheep by stimulating neutrophils against *M. hominis* and *M. agalactiae*, respectively [16,34,89]. Similarly, other authors failed to detect intact NET structures after stimulation of bovine neutrophils with living *M. bovis*, even in the presence of PMA; conversely, intact NET structures were visible by stimulating bovine neutrophils with heat-inactivated *M. bovis* cultures or with living *M. bovis* in which nucleases were neutralized by using EDTA as a chelating agent [36,105]. Taken together, data suggest that, as seen for other bacteria, mycoplasmas dismantle the NET’s DNA backbone by using their nucleases.

Membrane-associated or secreted nucleases were identified in at least 20 *Mycoplasma* species [107], but only a few of them were characterized (Table 1).

Nucleases are paramount to *Mycoplasma* pathogenesis. *Mycoplasma* nucleases digest host nucleic acids and provide free nucleotide precursors, promoting bacterial growth and persistence in the host [108]. Most of them do not show substrate selectivity and digest both RNA and DNA, either single or double-stranded. Moreover, most mycoplasma nucleases can also act as endonucleases and nick circular DNA, therefore fully digesting any type of free DNA (linear DNA, plasmids, and small circular genomes) by their exonuclease and endonuclease activities. The biochemical requirements for the activity of nucleases in different *Mycoplasma* species are similar and rely on the availability of divalent cations, such as Ca^2+^ and Mag^2+^. Interestingly, most of the described mycoplasma nucleases are annotated as homologs of the Staphylococcal nuclease (SNase), which was demonstrated as able to escape from NET entrapment and killing by digesting NET’s DNA backbone, reducing structure stability, and enhancing NET clearance [69].

**Table 1 ijms-23-15030-t001:** Mycoplasma nucleases characteristics.

Species	Nuclease Name	Subcellular Localization	Type of Nuclease	Substrate	Required Divalent Cation	References
*M. agalactiae*	MAG_5040	Membrane-exposed	Endo/Exo	ssDNA, dsDNA, RNA, and plasmid	Mg^2+^	[38]
*M. bovis*	MnuA	Membrane-exposed	Endo/Exo	dsDNA, plasmid,	Ca^2+^, Mg^2+^	[109]
	MbovNase	Membrane-bound and secreted	Endo	dsDNA, RNA, and plasmid	Ca^2+^	[110]
*M. hominis*	MHO_0730	Membrane-exposed	Endo/Exo	ssDNA, dsDNA, RNA, plasmid	Ca^2+^	[34]
*M. pneumoniae*	Mpn133	Membrane-associated	Endo/Exo	ssDNA, dsDNA, RNA, and plasmid	Ca^2+^	[111]
	Mpn491	Secreted	DNAse	DNA	Mg^2^	[37]
*M. genitalium*	MG_186	Membrane-associated	Endo/Exo	ssDNA, dsDNA, RNA, and plasmid	Ca^2+^	[112]
*M. hyopneumoniae*	Mhp379	Membrane-exposed	Endo/Exo	ssDNA, dsDNA, RNA, plasmid	Ca^2+^	[113]
	Mhp597	Secreted	Endo/Exo	ssDNA, dsDNA, RNA, and plasmid	Ca^2+^, Mg^2+^	[40]
*M. gallisepticum*	MGA_0676	Membrane-exposed	Endo/Exo	ssDNA, dsDNA, RNA, and plasmid	Ca^2+^	[114]
*M. meleagridis*	Mm19	Membrane-exposed	Endo/Exo	ssDNA, dsDNA, RNA, and plasmid	Mg^2^	[115]
*M. pulmonis*	MnuA	Membrane	Undefined	DNA	Undefined	[116]
*M. penetrans*	P40	Membrane	Endo/Exo	ssDNA, dsDNA, RNA, and plasmid	Ca^2+^, Mg^2+^	[117,118]

The difficulty to selectively interrupt mycoplasmas genes was hampered for a long time by the lack of full comprehension of the role of nucleases in the *Mycoplasma* life cycle. Consequently, few studies report the knock-out of *Mycoplasma* nuclease genes, such as *M. bovis, MnuA,* and *M. pneumoniae Mpn491*, interrupted using a traditional method based on transposons. In these studies, both wild-type and nuclease-deleted strains were able to efficiently induce NET release, but only the wild-type strain retained the ability to digest DNA and disassemble NETs [37,39,109]. Recently, the CRISPR/Cas system was adapted and successfully used to interrupt the gene encoding MGA_0637 in *M. gallisepticum*, paving the way for more specific studies in other mycoplasma species [119].

Surface nucleases, beyond NET escape, are fundamental to *Mycoplasma* metabolism. In fact, nuclease genes are usually included in ABC transporters operons, suggesting a role in providing free precursors for nucleic acids synthesis which can be internalized by the ABC transporter system proteins [38]. Additionally, by degrading macrophage extracellular traps, *M. hyopneumoniae* Mhp379 surface nuclease releases free nucleotides which are internalized into the mycoplasma cell and used to synthesize novel bacterial DNA [35].

Mycoplasma-induction of NET release has therefore a double sword effect: on the one hand, it promotes mycoplasma evasion of innate immune responses (phagocytosis, complement, and NETs), and on the other hand, it provides a virtually unlimited nucleotide supply, critical to mycoplasma persistence in the host considering their limited biosynthetic abilities.

### 3.1. Mycoplasma agalactiae

*M. agalactiae* is the etiological agent of contagious agalactia (reduction or failure of milk secretion) sensu stricto, a detrimental disease of small ruminants causing severe economic losses worldwide [120]. *M. agalactiae* is able to establish chronic infections, causing recurrent symptoms such as mastitis, arthritis, keratoconjunctivitis, and occasionally abortion [120]. The persistence of *M. agalactiae* shedding into milk was reported up to 8 years after infection, independently from the presence of clinical signs, thus promoting pathogen dissemination into the environment and spreading infection [121,122].

Virulence factors related to the chronicization and persistence of *M. agalactiae* infection are still not sufficiently defined. The combination of full genome sequencing and gene ontology analyses suggests that *M. agalactiae* pathogenicity does not rely on primary virulence factors, such as toxins, invasins, and cytolysins. A direct role in pathogenicity was proposed only for a few proteins, mostly involved in adhesion to the host cell [123,124,125,126,127]. Proteomic studies identified proteins associated with the *M. agalactiae* membrane and antigens eliciting humoral response along natural infection [94,128]. Among them, a SNase domain was identified in the surface lipoprotein MAG_5040 [38].

*M. agalactiae* MAG_5040 is a broad-range nuclease, able to digest dsDNA (both linear and circular), ssDNA, and RNA. Moreover, MAG_5040 is an antigen able to elicit a small ruminant humoral response during natural infection; specific antibodies appear in the early stages of infection and persist at least for 9 months [38,128]. The biochemical characteristics of MAG_5040 are reported in Table 1.

MAG_5040 is a thermostable enzyme and is active up to 65 °C, with peak activity between 37 °C and 45 °C; this characteristic probably promotes *M. agalactiae* survival in different body regions, such as the poorly thermoregulated conjunctiva and the mammary gland, where the temperature ranges from 38 to 40 °C under physiological conditions and could rise during inflammation.

The gene encoding MAG_5040 is located upstream of an ABC transporter, in a genetic organization that can be observed in several mycoplasma species [38]. The MAG_5040 genomic context and 3D modeling support the hypothesis that this protein and the ABC transporter are part of a machinery involved in the uptake of nucleotides, in which MAG_5040 nuclease plays a predominant role by digesting the host nucleic acids [38,123].

The involvement of MAG_5040 was also speculated in NET dismantling during natural infection [33]. The production of a MAG_5040 deficient strain still represents a necessary step for the explicit recognition of the role of MAG_5040 in NET evasion.

### 3.2. Mycoplasma bovis

*M. bovis* infection is linked to a variety of syndromes in cattle, such as respiratory disease, mastitis, conjunctivitis, arthritis, and otitis media, representing a serious threat and causing severe economic losses in beef and dairy cattle worldwide [129]. Fragmentary information is available on the molecular and pathogenetic mechanisms underlying *M. bovis* infection. As seen in other mycoplasma species, it is well assessed that the membrane proteins contribute to this process, mainly modulating host immune response [130].

*M. bovis’* ability to induce NETs was debated until 2018 [36].

Two different nucleases were described in *M. bovis* (MnuA and MbovNase). While the major membrane nuclease MnuA is surface exposed, the MbovNase has a double localization, being both membrane-associated and secreted. Both of these enzymes are able to digest a wide range of substrates [39,109,110].

Despite recombinant MbovNase being able to digest pre-formed NETs in vitro, the disruption of the correspondent gene did not significantly affect *M. bovis’* ability to dismantle NETs ex vivo [109,110]. Conversely, knocking out MnuA abolishes most *M. bovis* nuclease activity, and the MnuA defective strain loses the ability to dismantle NETs ex vivo [109]. Considering the different subcellular locations of MnuA and MbovNase, it was speculated that MnuA nuclease activity is mainly restricted to the cell surface, while it is feasible that MbovNase is less relevant in NET dismantling. In turn, MbovNase is a cytotoxic secreted protein that is able to invade BoMac cells in vitro, inducing the apoptosis of invaded cells [110].

Oddly, the gene encoding for the MbovNase (MBOV_RS02825) is associated with a nucleotide-ABC transporter operon, as seen in *M. agalactiae*, while MnuA is associated with several other enzyme-encoding genes not related to transport systems [38,131].

More in vivo studies are needed to elucidate the mechanisms of induction and dismantling of NETs during *M. bovis* infection, although it is plausible that the general picture is very similar to what was demonstrated for *M. agalactiae* [16,89].

### 3.3. Mycoplasma hominis

*M. hominis* is an opportunistic pathogen of the human urogenital tract, associated with a wide range of diseases, such as pelvic inflammation, urethritis, and pregnancy and postpartum complications; moreover, if it gains access, *M. hominis* can establish infections, especially in immunocompromised patients, in other body regions, such as pericardium, joints, ear, brain, and meninges, and can be involved in prostate cancer development [132,133,134,135,136,137,138,139]. *M. hominis* is also able to invade and establish a symbiotic relationship with *T. vaginalis* [140].

As seen in other mycoplasma species, the pathogenicity mechanisms underlying *M. hominis* infections still need to be elucidated and only a few virulence factors have been identified.

*M. hominis* triggers NET release through its surface lipoproteins [16]. When neutrophils are co-incubated with *M. hominis*, intact NET structures are detected only when bacteria are pre-treated with paraformaldehyde, and not when viable mycoplasmas are used, indicating the presence of mycoplasma factors able to digest NETs [34].

*M. hominis* expresses MHO_0730, a homologue of the Staphylococcal SNase. MHO_0730 is a surface-anchored lipoprotein, acting as a nuclease with its amino acidic portion, and able to trigger NET release thanks to its N-terminal lipid moiety, either in the Pam2Cys or Pam3Cys versions [34,141]. As seen in *M. agalactiae*, MHO_0730 is a thermostable enzyme that digests several nucleic acid substrates and is also able to elicit the host humoral response [34]. The MHO_0730 gene is located upstream of an operon encoding for an ABC transporter, as seen in other mycoplasmas [38,142].

The role of MHO_0730 in the symbiotic relationship between *M. hominis* and *T. vaginalis* is still unclear and more studies are needed to elucidate the significance of this protein and its potential ability to promote *T. vaginalis* NET evasion.

### 3.4. Mycoplasma pneumoniae

*M. pneumoniae* is a human pathogen causing acute and chronic respiratory infections, usually causing mild symptoms that may evolve into pneumonia, though it can be associated with more severe diseases and extrapulmonary manifestations involving joints, the kidney, pancreas, liver, skin, and cardiovascular and central nervous system. Using several mechanisms, *M. pneumoniae* can adhere to (and invade) airway cells, damaging epithelium integrity, decreasing ciliary movement, and causing vacuolation and exfoliation [143].

*M. pneumoniae* viability is not affected by neutrophils, even though an increase in neutrophil alveolar infiltration during mycoplasma-induced pneumonia is observed [144]. Indeed, *M. pneumoniae* has evolved several strategies to evade neutrophil clearance by avoiding both neutrophils phagocytosis and NET extracellular killing [145,146].

*M. pneumoniae* induces the release of NETs ex vivo, and subsequently degrades their DNA structure; in vivo, neutrophil recruitment is coupled with an increase in IL-8 in the bronchoalveolar fluids [144].

Among the proteins involved in *M. pneumoniae* infection, Mpn491 and Mpn133 exhibit nuclease activity [37,111].

Mpn491 is a secreted nuclease and represents the major contributor to *M. pneumoniae’s* resistance to NETs-mediated killing. Indeed, Mpn491 knockout impairs NETs degradation by the mutant strain, both ex-vivo and in vivo, in mouse models [37]. At a genomic level, Mpn491 is not associated with an ABC transporter encoding operon [147].

Mpn133 is a thermostable nuclease with well-assessed cytotoxic properties. This enzyme is able to digest a wide range of nucleic acids, showing both exo- and endo-nuclease activities. Since the Mpn133 encoding gene is located upstream of an ABC transporter operon, it can be postulated that Mpn133 has a role in the nucleotide uptake from the host [38,108,111,147]. In addition to the SNase domain, Mpn133 has a unique glutamic acid, lysine, and serine-rich region (EKS region), essential for the induction of cytopathic effects. Indeed, the presence of this motif allows the internalization of Mpn133 in human airway A549 cells and the translocation to the nucleus; here, Mpn133 digests the host DNA, decreasing the cell viability through an apoptosis-like death [111]. The involvement of Mpn133 in the NET degradation was not investigated, but its contribution cannot be excluded.

Mpn491 and Mpn133 nucleases represent crucial virulence determinants for *M. pneumoniae*, providing different methods of access to the host cell nucleotide pools, and allowing escape from the host’s innate immune system.

### 3.5. Mycoplasma hyopneumoniae

*M. hyopneumoniae* is a major threat to pig production, causing complex diseases of the respiratory tract. *M. hyopneumoniae* infects the trachea ciliated epithelium, bronchi, and bronchiole, damaging the cilia and causing cell death. Due to its ability to modulate and evade the host immune response and to reduce the clearance efficiency of mucociliary apparatus, secondary infections occur frequently, causing either enzootic pneumonia or porcine respiratory disease complex [148].

*M. hyopneumoniae* is able to induce NET release and dismantling [40]. The main nuclease involved in NET digestion is Mhp597, a secreted enzyme detectable in the active form in *M. hyopneumoniae* culture supernatant. Mhp597 has also cytotoxic and pro-inflammatory properties. Indeed, recombinant Mhp597 (rMhp597) induces apoptosis in PK15 cells in a dose-dependent manner. Moreover, rMhp597 causes alterations in the expression of immunity and inflammation-related factors in porcine alveolar macrophages ex vivo; more specifically, rMhp597 causes the down-regulation of IFN-α/β and upregulation of IL-1β, IL-8, and TNF-α [40].

Mhp379 was the first nuclease identified in *M. hyopneumoniae* [113]. Similar to what is observed for most of the mycoplasma nucleases, both Mhp597 and Mhp379 are able to digest both DNA and RNA and display exo-and endo-nuclease activity, being able to nick closed plasmid DNA [40,113]. Despite the localization on the cell surface, to date, the involvement of Mhp379 in NETs escape has not been investigated.

The genomic context of the genes encoding for the two nucleases is extremely different. The Mhp379 gene is located upstream of a conserved ABC transporter operon, and it can be speculated to have an association between Mhp379 and the transport system [113]. Otherwise, Mhp597 is associated with several genes encoding for enzymes involved in DNA replication and the initiation complex for protein synthesis [149].

## 4. Relevance of NETs to Immunity and Mycoplasma Acute and Chronic Infections

Although mycoplasmas are associated with illness in several species, the multitude of pathways involved in pathogenesis are not fully understood, both on the microbial and host sides. In particular, the role of host innate immunity needs to be investigated more thoroughly.

Upon infection, mycoplasma PAMPs, mainly represented by lipoproteins, trigger the release of proinflammatory cytokines and chemokines by epithelial cells and leukocytes, leading to the recruitment of granulocytes, macrophages, and lymphocytes to the site of infection [30]. Among the activated pathways, a paramount role is accomplished by neutrophils, with the release of ROS, NETs, and IL-8, creating an inflammatory milieu that can result in tissue damage, as seen in mycoplasma-related mastitis [16,30,33,150].

Mycoplasmas induce an acute inflammatory response, but chronic infections with recurrent symptoms are frequent. Persistent *Mycoplasma* infections, often asymptomatic, can also be associated with post-infection inflammatory diseases, such as chronic respiratory diseases (atypical pneumonia and asthma), mastitis, pelvic and urogenital inflammatory pathologies, and autoimmune and immunosuppressive diseases [21,122,130,138,143,148,151,152]. Fragmentary data about the involvement of mycoplasmas in the development of these diseases are available for human infections. Studies focus on epidemiological investigation, while the mechanisms and actors promoting chronic inflammation and damage are poorly investigated.

Among factors triggering an immune response, a key player is represented by the interaction of mycoplasma PAMPs with the TLRs [30]. In light of this, mycoplasma lipoproteins are likely involved in the onset and development of systemic immune-mediated diseases, at least by interacting with TLR1/2 or TLR2/6 and activating NET formation [16,34]. Serological investigations have demonstrated that sera collected from patients with rheumatoid arthritis, but not with systemic lupus erythematosus, have antibodies against a higher number of *M. hominis* and *M. fermentans* lipoproteins compared to healthy control sera [153]. These data suggest a contribution of mycoplasma lipoproteins to disease development and/or symptom exacerbation. Since lipoproteins are the major triggers for mycoplasma-induced NET release, a possible role of NET release in pathogenesis could be speculated [16].

An increased presence of *Mycoplasma* spp. was observed in the gut microbiota of patients with inflammatory bowel and Crohn’s diseases, in conjunction with an inflammatory context, with an altered cytokine profile, suggesting a role in the development of these syndromes [154,155].

In addition to the respiratory tract, *M. pneumoniae* can colonize the musculoskeletal, nervous, hematological, digestive, and renal systems, causing tissue damage and triggering immune cell recruitment and inflammatory responses. *M. pneumoniae* infection is associated with an increased risk of systemic lupus erythematosus and rheumatoid arthritis [156]. However, few mechanisms are described. Among them, *M. pneumoniae* is able to induce low C3-complement immune-mediated injury by deposition of immune complexes in the urinary tract, leading to nephritis, which is one of the manifestations of systemic lupus erythematosus [157,158].

The involvement of *M. hominis* in septic arthritis was also reported in patients with systemic lupus erythematosus or rheumatoid arthritis [159,160]; a link between *M. fermentans* and arthritis was also suggested [161].

The persistence of mycoplasma infection and the consequent chronic inflammation leads to a progressive malignant transformation, increasing the in vitro invasiveness and in vivo metastasis of different human tumor cells [162]. Epidemiologic studies demonstrated the association of several mycoplasma species with different kinds of tumors [163]. *M. hominis*, *Ureaplasma parvum*, and *Ureaplasma urealyticum* are tightly associated with prostate cancer [136,162,164,165]. Moreover, *M. hominis* and *M. genitalium* infections are associated with an increased risk of cervical cancer [166,167]. *M. fermentans* or *M. penetrans* induced transformation in C3H mouse embryo cells through a multistage process producing prominent chromosomal changes [168].

Many pathways are involved in mycoplasma-driven transformation, and the constant inflammatory status has a pivotal role. It was demonstrated that *M. hyorhinis* induces the malignant transformation of gastric cells through different pathways. Among them, the activation of the NLRP3 inflammasome, which induces the maturation of proinflammatory cytokines, has a prominent role [169].

*Mycoplasma* can contribute to the onset of inflammatory disease and tumors through several mechanisms and involving different virulence factors, but the role of NETs cannot be underrated. To date, few fragmentary data are available, and more specific studies must be conducted to clarify the contribution of NETs to septic inflammatory diseases.

## 5. Conclusions

*Mycoplasma* infections modulate innate and adaptive immune responses, producing acute and chronic diseases in both animals and humans. The initial interaction of mycoplasmas with inflammatory cells, such as neutrophils, is paramount to progressive events leading to mycoplasma survival and disease progression.

In addition to reducing the efficacy of phagocytosis, NETs’ induction and disassembling allow mycoplasmas to counteract early bacteria clearance and at the same time provide nucleotides essential to mycoplasma replication and survival by complementing the absence of nucleotide de novo synthesis. By using their potent nucleases, mycoplasmas are therefore able to turn neutrophil cells into a source of essential food (nucleotides), basically by “eating the enemy” (Figure 1). This is a key mechanism in mycoplasma long-term cell survival which promotes the establishment of persistent infections and possibly contributes to many inflammatory chronic diseases.

## Figures and Tables

**Figure 1 ijms-23-15030-f001:**
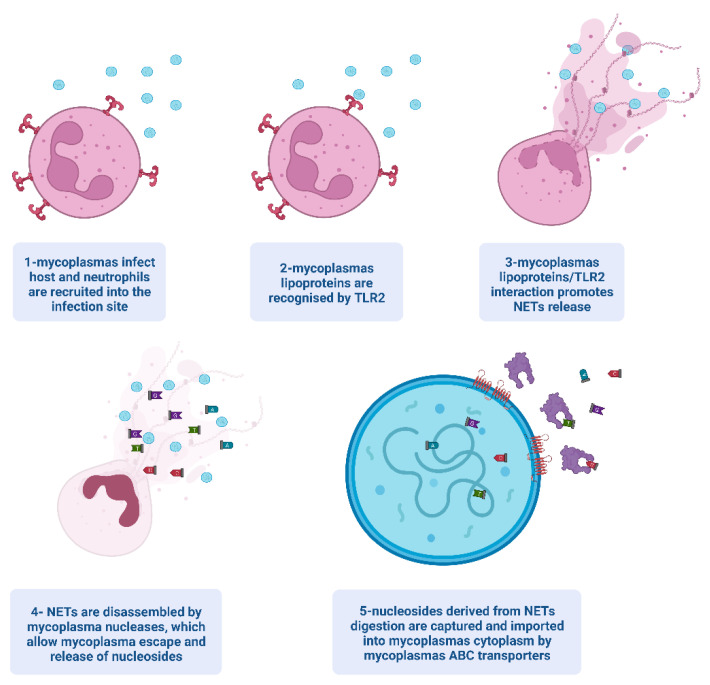
Mechanism of NET induction and disassembling and the mycoplasma uptake of generated nucleotides. Created with BioRender.com.

## Data Availability

Data sharing not applicable.

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
