# Peer review of "Eating the Enemy: Mycoplasma Strategies to Evade Neutrophil Extracellular Traps (NETs) Promoting Bacterial Nucleotides Uptake and Inflammatory Damage"

_ijms, 2022, doi:10.3390/ijms232315030_

Round 1

Reviewer 1 Report

This paper is of great scientific value because it is a broad and comprehensive overview of NETs. It is very interesting not only to show the involvement of NETs in pathogenesis, but also to mention the individual characteristics of many Mycoplasma spcies.

I am convinced that this is a very valuable review that will lead to the elucidation of the pathogenesis of infections caused by Mycoplasma species and the construction of methods for their control in the future.

It is a carefully scrutinised review and we have nothing to point out.

Author Response

We thank this reviewer for the positive feedback.

Reviewer 2 Report

This is a comprehensive and thorough review of the interactions of mycoplasma and neutrophil extracellular trap (NET) formation and persistence. The authors have done a very thorough job of covering the area, both broadly and specifically in relation to a number of particular mycoplasma strains with disease associations.

Overall it is well written. It is apparent that the manuscript is written by non-native English speakers and as such it could be improved by editing by a native English speaker. For example, there are numerous examples of using plural form when a singular form is appropriate. Whether this should be done by the authors or at the editorial/typesetting stage is unclear

Major

The acronym ‘AMPs’ is used but not defined. I assume it means anti-microbial proteins, but please clarify and define.

Line 126 – explain what is meant by ‘dynamic conditions.’ Is this relating to blood flow?

Line 157 – Provide a reference supporting the statement re NET removal by C1q. I have not heard of this before but I have heard of a role for C1q in removal of apoptotic cells.

Line 187/189 – ‘Dissemination of metastasis’ is a tautology. Metastasis alone or 'dissemination of cancer cells' would be better.

Lines 255-257 and beyond – For clarity it would help the reader to compare and contrast the activities of these bacterial nucleases with mammalian DNAses. Explain what their similarities and differences are.

Line 296 – define agalactia for non-specialist audience

Minor

Replace all examples of ‘associated to’ with ‘associated with’.

Replace instances of ‘composed by’ with ‘composed of’.

Line 76 – replace ‘are’ with ‘being’

Line 229 – ‘allowed to identify’ – clarify – ‘allowed identification of’?

Line 263 – double not doble

Line 265 – explain difference between exo and endo nuclease

Line 416 – threat not treat

Author Response

Review 2

This is a comprehensive and thorough review of the interactions of mycoplasma and neutrophil extracellular trap (NET) formation and persistence. The authors have done a very thorough job of covering the area, both broadly and specifically in relation to a number of particular mycoplasma strains with disease associations.

Overall it is well written. It is apparent that the manuscript is written by non-native English speakers and as such it could be improved by editing by a native English speaker. For example, there are numerous examples of using plural form when a singular form is appropriate. Whether this should be done by the authors or at the editorial/typesetting stage is unclear

We thank the reviewer for this important comment. The article has been extensively reviewed in terms of general use of English.

Major

The acronym ‘AMPs’ is used but not defined. I assume it means anti-microbial proteins, but please clarify and define.

The acronym AMPs is now defined in the manuscript (Due to massive presence of antimicrobial peptides (AMPs), it….).

Line 126 – explain what is meant by ‘dynamic conditions.’ Is this relating to blood flow?

This sentence was modified into: “It was observed that NETs generated in static conditions (such as by adherent neutrophils) have fewer killing abilities than NETs released under dynamic conditions (non-adherent neutrophils)”.

Line 157 – Provide a reference supporting the statement re NET removal by C1q. I have not heard of this before but I have heard of a role for C1q in removal of apoptotic cells.

A reference (77 in the reference section) was provided at the end of the sentence, according to journal rules for reference citations in the text.

Line 187/189 – ‘Dissemination of metastasis’ is a tautology. Metastasis alone or 'dissemination of cancer cells' would be better.

That is really tautological, we are sorry we did not really realize that while writing, now we use ‘metastasis’

Lines 255-257 and beyond – For clarity it would help the reader to compare and contrast the activities of these bacterial nucleases with mammalian DNAses. Explain what their similarities and differences are.

Considering that we provided a detailed table about characteristics of mycoplasma nucleases we would rather avoid the use of additional editorial space, but if the reviewer consider this mandatory we are open to modify the manuscript.

Line 296 – define agalactia for non-specialist audience

The sentence now reads: “…of contagious agalactia (reduction or failure of milk secretion)…”

Minor

Replace all examples of ‘associated to’ with ‘associated with’.

done

Replace instances of ‘composed by’ with ‘composed of’.

done

Line 76 – replace ‘are’ with ‘being’

done

Line 229 – ‘allowed to identify’ – clarify – ‘allowed identification of’?

This sentence has been modified into: “Recently, mass spectrometry allowed the identification of a triacylated”.

Line 263 – double not doble

done

Line 265 – explain difference between exo and endo nuclease

We would rather skip this, if possible

Line 416 – threat not treat

done

Round 2

Reviewer 2 Report

I am happy to accept this manuscript following these changes.